# Osirisynes G-I, New Long-Chain Highly Oxygenated Polyacetylenes from the Mayotte Marine Sponge *Haliclona* sp.

**DOI:** 10.3390/md18070350

**Published:** 2020-07-03

**Authors:** Pierre-Eric Campos, Gaëtan Herbette, Christophe Chendo, Patricia Clerc, Florent Tintillier, Nicole J. de Voogd, Eleni-Dimitra Papanagnou, Ioannis P. Trougakos, Moran Jerabek, Jérôme Bignon, Géraldine Le Goff, Jamal Ouazzani, Anne Gauvin-Bialecki

**Affiliations:** 1Laboratoire de chimie et de biotechnologie des produits naturels, Faculté des Sciences et Technologies, Université de La Réunion, 15 Avenue René Cassin, CS 92003, 97744 Saint-Denis CEDEX 9, La Réunion, France; pierre-eric.campos@univ-reunion.fr (P.-E.C.); patricia.clerc@univ-reunion.fr (P.C.); florent.tintillier@gmail.com (F.T.); 2Aix-Marseille Univ, CNRS, Centrale Marseille, FSCM, Spectropole, Campus de St Jérôme-Service 511, 13397 Marseille, France; gaetan.herbette@univ-amu.fr (G.H.); christope.chendo@univ-amu.fr (C.C.); 3Naturalis Biodiversity Center, Darwinweg 2, 2333 CR Leiden, The Netherlands; nicole.devoogd@naturalis.nl; 4Institute of Environmental Sciences, Leiden University, Einsteinweg 2, 2333 CC Leiden, The Netherlands; 5Department of Cell Biology and Biophysics, Faculty of Biology, National and Kapodistrian University of Athens, Athens 15784, Greece; epapanagnou@biol.uoa.gr (E.-D.P.); itrougakos@biol.uoa.gr (I.P.T.); 6Crelux GmbH, Am Klopferspitz 19a, 82152 Planegg-Martinsried, Germany; Moran_Jerabek@wuxiapptec.com; 7Institut de Chimie des Substances Naturelles, CNRS UPR 2301, Université Paris-Sud, Université Paris-Saclay, 1, av. de la Terrasse, 91198 Gif-sur-Yvette, France; jerome.bignon@cnrs.fr (J.B.); geraldine.legoff@cnrs.fr (G.L.G.); Jamal.Ouazzani@cnrs.fr (J.O.)

**Keywords:** *Haliclona* sp., marine sponge, oxygenated polyacetylenes, enzyme inhibitors

## Abstract

Chemical study of the CH2Cl2−MeOH (1:1) extract from the sponge *Haliclona* sp. collected in Mayotte highlighted three new long-chain highly oxygenated polyacetylenes, osirisynes G-I (**1**–**3**) together with the known osirisynes A (**4**), B (**5**), and E (**6**). Their structures were elucidated by 1D and 2D NMR spectra and HRESIMS and MS/MS data. All compounds were evaluated on catalase and sirtuin 1 activation and on CDK7, proteasome, Fyn kinase, tyrosinase, and elastase inhibition. Five compounds (**1**; **3**–**6**) inhibited proteasome kinase and two compounds (**5**–**6**) inhibited CDK7 and Fyn kinase. Osirisyne B (**5**) was the most active compound with IC_50_ on FYNB kinase, CDK7 kinase, and proteasome inhibition of 18.44 µM, 9.13 µM, and 0.26 µM, respectively.

## 1. Introduction

Natural products play a highly significant role as a source of new and approved drugs for the treatment of human diseases. Approximately 70% of small molecule drugs were produced between 1981 and 2006, possessing an important link to a natural product source [1,2]. Over the last few decades, the oceans (covering 70% of the earth) have represented a widely promising source of new biologically active natural compounds [3] with considerably different characteristics in comparison with those of the terrestrial ones [4].

Among natural compounds, polyacetylenes are widely distributed, occurring in plants, moss and lichens, fungi, marine algae, sponges, tunicates, insects, and frogs [5]. More specifically, in the phylum Porifera, the main source of long-chain polyacetylenes with polyketide or fatty acid origin is marine sponges of the order Haplosclerida including genera belonging to different families, namely *Petrosia* [6,7,8], *Xestospongia* [9] (Petrosiidae), *Cribochalina* [10] (Niphatidae), *Haliclona* [11,12] (Chalinidae), *Siphonochalina* [13], and *Callyspongia* [14] (Callyspongiidae). Some of these compounds are known to possess potent bioactivities such as antimicrobial [11], antiviral [6], antifungal [7], cytotoxic [10], and enzyme-inhibitory activities [12]. They have also been considered useful as nutraceuticals for the development of healthier foods [15].

In health, enzymes play key roles in different cellular processes and their deregulation has been considered as one of the first causes of age-related diseases, including cancer [16,17] and Alzheimer’s disease [18,19]. As good drug candidates, natural enzyme activators or inhibitors have received an increasing amount of attention for their potential pharmacological applications, especially those from marine origin [20].

In our continuing search for bioactive metabolites from marine invertebrates, the undescribed sponge *Haliclona* sp. collected in Mayotte (Indian Ocean), was investigated. The organic crude extract of this animal exhibited a potent inhibitory activity against proteasome as well as a significant inhibitory activity against tyrosinase. Bioassay-guided partitioning and separation by chromatographic methods led to the isolation of the three known long-chain highly oxygenated polyacetylenes osirisynes A (**4**), B (**5**), and E (**6**) together with three new long-chain highly oxygenated polyacetylenes osirisynes G-I (**1**–**3**) (Figure 1). The purification and structure elucidation by spectral data including HRESIMS, MS/MS, and 2D NMR and in comparison with published data [12], are reported herein. The biological evaluations of the latter new compounds against seven different targets involved in aging or age-related diseases are described as well.

## 2. Results and Discussion

### 2.1. Chemistry

The CH_2_Cl_2_-MeOH extract of the lyophilized sponge *Haliclona* sp. was first subjected to a normal-phase silica gel column chromatography to yield 12 fractions. Fraction 9 was subjected to repetitive reverse-phase semi-preparative and analytical HPLC to yield six pure compounds (**1**–**6**) (Figure 2). Among them, three are known and were identified as osirisynes A (**4**), B (**5**), and E (**6**) by comparison with published spectroscopic data; the other three are new and were named osirisynes G-I (**1**–**3**). Their elucidation is described below.

Osirisyne G (**1**) was obtained as a white amorphous solid. The molecular formula, C_47_H_72_O_12_, was established from a HRESIMS molecular ion peak at *m*/*z* 827.4950 [M − H]^−^, indicating 12 degrees of unsaturation (Appendix A). Analysis of the 1D and 2D ^1^H and ^13^C NMR data for **1** (CD_3_OD, Table 1, Appendix A) revealed resonances and correlations consistent with those of a long-chain highly oxygenated polyacetylene, like osirisynes A–F [12] or fulvynes A–I [11]. The ^1^H NMR spectrum of **1** recorded in CD_3_OD showed the presence of four olefinic protons (δ_H_ 5.88 (1H, ddd, *J* = 15.4, 6.2, 1.3 Hz), δ_H_ 5.76 (1H, ddd, *J* = 15.4, 5.7, 1.1 Hz), δ_H_ 5.62 (1H, dtd, *J* = 15.3, 6.5, 0.8 Hz), and δ_H_ 5.43 (1H, ddt, *J* = 15.3, 7.1, 1.4 Hz), an acetylenic proton [δ_H_ 2.92 (1H, d, *J* = 2.2 Hz), nine oxygenated methines (δ_H_ 5.11 (1H, m), δ_H_ 4.82 (1H, dm, *J* = 5.7 Hz), δ_H_ 4.60 (1H, d, *J* = 4.2 Hz), δ_H_ 4.33 (1H, td, *J* = 6.7, 1.6 Hz), δ_H_ 4.09 (1H, q, *J* = 6.0 Hz), δ_H_ 3.97 (1H, m), δ_H_ 3.69 (1H, tt, *J* = 10.9, 6.3 Hz), δ_H_ 3.61 (1H, td, *J* = 8.6, 2.5 Hz), and δ_H_ 3.43 (1H, dd, *J* = 8.1, 4.3 Hz) and a series of methylene groups in the range δ_H_ 2.50–1.30. The ^13^C NMR spectrum of **1** showed the presence of a ketone C-19 (δ_C_ 214.5), a carboxylic acid C-1 (δ_C_ 161.3), eight sp carbons due to four triple bonds C-2, C-3, C-32, C-33, C-35, C-36, C-46, and C-47 (δ_C_ 79.9, 83.4, 85.0, 83.8, 80.9, 81.9, 84.7, 74.9), four sp^2^ carbons due to two double bonds C-25, C-26, C-43, and C-44 (δ_C_ 132.3, 134.3, 136.0, 130.3), nine oxymethines C-4, C-5, C-6, C-27, C-31, C-34, C-38, C-42, and C-45 (δ_C_ 65.0, 78.5, 72.7, 73.5, 62.5, 52.3, 70.7, 72.2, 62.5), and several methylene groups were also present.

All proton-bearing carbons were assigned by HSQC experiment. Analysis of the COSY and HMBC correlations aided in recognizing the partial structures a–d (Figure 3) of the long alkyl chain of compound **1**. The COSY correlations revealed the presence of the spin system C-4—C-5—C-6—C-7 for the partial structure a and the HMBC correlations between H-4, C-2, and C-3 and between H-5 and C-2 indicated the carbon resonances of the triple bond in fragment a. For the partial structure b, the HMBC correlations between H-18 and C-19, H-20 and C-19, C-21 and C-22, H-21 and C-19, C-22 and C-23, in addition to the COSY correlations between H-17 and H-18 as well as H-20 and H-21 indicated the presence of the ketone C-19 in the spin system C-17—C-18—C-19—C-20—C-21—C-22. The COSY correlations between H-24 and H-25, H-25 and H-26, H-26 and H-27, as well as H-27 and H-28 revealed the presence of an oxymethine C-27 linked to a sp^2^ carbon C-26 in the spin systems C-24—C-25—C-26—C-27—C-28. The two spin systems mentioned above can be linked by analysis of HMBC correlations between H-22, C-23, and C-24; H-23, C-24, and C-25; H-24, C-25, and C-26; H-25, C-23, H-26, and C-24. HMBC correlations between H-31, C-32, and C-33; H-34 and C-32; C-33, C-35, and C-36; and H-37, C-34, C-3,5 and C-36 indicated the carbon resonances of the two triple bonds in fragment c, thus, by using the COSY correlations, the spin system C-30—C-31—C-32—C-33—C-34—C-35—C-36—C-37—C-38—C-39 was revealed. The partial structure d was revealed by COSY correlations between H-41 and H-42; H-42 and H-43; H-43 and H-44; H-44 and H-45; and between H-45 and H-47 and by HMBC correlations between H-45 and C-43, C-44, C-46, and C-47 and between H-44 and C-42 and C-46. The geometries of the two double bonds were easily assigned as 25*E*, 43*E* by the coupling constant analysis of the olefinic protons (*J* = 15.3 and 15.4 Hz, respectively). The relative configuration of compound **1** remained unassigned. Any attempts to obtain suitable derivatives (Mosher’s esters) for a stereochemical analysis were unsuccessful due to the small amount of product and the high number of chiral carbons. The connectivities between partial structures a–d for **1** as well as the number of the linking methylene groups were established on the basis of the molecular formula and ESI–MS/MS data. The combinations of partial structures a + b + c + d represented 674 m.u. whereas the molecular structure weight was 828 m.u. The difference corresponding to 11 methylene groups determined the length of the alkyl chains between the different partial structures. For the ESI-MS/MS analysis, Cu^II^ was used for compound ionization so the species sought were detected as [(M − H) + Cu^II^]^+^. Indeed, a reduction of Cu^II^ copper to Cu^I^ can occur during the electrospray ionization process [21]. This reduction is accompanied by the formation of radical species, which can be at the origin of specific fragmentations during the experiments of collision-induced dissociation (CID) and cannot be obtained from the dissociation of protonated species [M + H]^+^ for this type of compound [22]. Indeed, the CID of these protonated species is generally accompanied by the loss of non-specific molecules such as H_2_O molecules. The ESI–MS/MS spectra (Figure 4) showed different fragment ions that indicated the presence of nine methylenes between a and b, one methylene between b and c, and one methylene between c and d. The fragmentation of the molecule was explained by different dissociation mechanisms (Figure 5). In ESI^+^–MS/MS, the peak at *m*/*z* 846.4 corresponded to the loss of a CO_2_ molecule and confirmed the presence of the carboxylic acid C-1. The presence of this carboxylic acid was also confirmed in ESI^−^–MS/MS by the presence of the ion [M − H]^−^ at *m*/*z* 827.5 and the peak at *m*/*z* 783.5 corresponded to the loss of a CO_2_ molecule. In ESI^+^–MS/MS, the peak at *m*/*z* 761.4 can be explained by a rearrangement between the three hydroxyls in C-4, C-5, and C-6, yielding to a loss of a H_2_O molecule and a 129.0 Da fragment corresponding to the C_5_H_3_O_3_ formula. This mechanism confirmed the sequence C-1 to C-6. The peak at *m*/*z* 542.2 can be explained by a fragmentation between C-27 and C-28. This fragment confirmed the link between the partial structure b and c by one methylene. The peak at *m*/*z* 708.3 can be explained by a fragmentation between C-37 and C-38. The loss of a 182.1 Da fragment corresponding to a C_10_H_14_O_3_ confirmed the link between the partial structure c and d by one methylene. In ESI^−^–MS/MS, the peak at *m*/*z* 601.4 was also due to a fragmentation between C-37 and C-38 and confirmed this link. These fragments (*m*/*z* 542.2 and 708.3 in ESI^+^–MS/MS and *m*/*z* 601.4 in ESI^−^–MS/MS) established the link between partial structures b, c, and d. In order to correspond to the molecular formula C_47_H_72_O_12_, the aliphatic chain between the partial structures a and b had to possess 12 methylenes.

Osirisyne H (**2**) was obtained as a white amorphous solid. The molecular formula, C_47_H_72_O_11_, was established from a HRESIMS molecular ion peak at *m*/*z* 811.5002 [M − H]^−^, indicating 12 degrees of unsaturation (Appendix A). Analysis of the 1D and 2D ^1^H and ^13^C NMR data for **2** (Table 2, Appendix A) revealed resonances and correlations consistent with those of a long-chain highly oxygenated polyacetylene, like osirisyne G (**1**). The ^1^H NMR spectrum of **2** recorded in CD_3_OD showed the presence of four olefinic protons (δ_H_ 5.88 (1H, ddd, *J* = 15.4, 6.2, 1.3 Hz), δ_H_ 5.76 (1H, ddd, *J* = 15.4, 5.7, 1.1 Hz), δ_H_ 5.61 (1H, dtd, *J* = 15.3, 6.7, 0.6 Hz), and δ_H_ 5.42 (1H, ddt, *J* = 15.3, 7.1, 1.3 Hz), an acetylenic proton (δ_H_ 2.92 (1H, d, *J* = 2.2 Hz), eight oxygenated methines (δ_H_ 5.04 (1H, quint, *J* = 1.8 Hz), δ_H_ 4.82 (1H, dm, *J* = 5.7 Hz), δ_H_ 4.60 (1H, d, *J* = 4.3 Hz), δ_H_ 4.09 (1H, q, *J* = 6.1 Hz), δ_H_ 3.97 (1H, q, *J* = 6.3 Hz), δ_H_ 3.69 (1H, m), δ_H_ 3.62 (1H, td, *J* = 8.8, 2.3 Hz), and δ_H_ 3.43 (1H, dd, *J* = 8.1, 4.3 Hz), and a series of methylene groups in the range δ_H_ 2.50–1.30. The ^13^C NMR spectrum of **2** showed the presence of a ketone C-19 (δ_C_ 214.5), a carboxylic acid C-1 (δ_C_ 158.6), eight sp carbons due to four triple bonds C-2, C-3, C-32, C-33, C-35, C-36, C-46, and C-47 (δ_C_ 80.0, 83.3, 79.3, 84.3, 81.1, 81.5, 84.0, 74.5), four sp^2^ carbons due to two double bonds C-25, C-26, C-43, and C-44 (δ_C_ 132.2, 134.3, 136.0, 130.4), eight oxymethines C-4, C-5, C-6, C-27, C-34, C-38, C-42, and C-45 (δ_C_ 65.0, 78.5, 72.9, 73.4, 52.4, 70.6, 72.2, 62.4), and several methylene groups were also present. Osirisyne H (**2**) was different from **1** by the presence of the methylene C-31 (δ_H_ 2.23 (2H, td, *J* = 6.9, 2.0 Hz); δ_C_ 19.1) instead of an oxygenated methine. The molecular formula determined by HRESIMS, C_47_H_72_O_11_ for osirisyne H (**2**) and C_47_H_72_O_12_ for osirisyne G (**1**), in addition to the HSQC correlation between H-31 and C-31 and the HMBC correlations between H-31, C-32, and C-33, confirmed this difference. The connectivities between partial structures a, b, c’, and d for **2** as well as the number of the linking methylene groups were established on the basis of the molecular formula and ESI–MS/MS data. As osirisyne G (**1**), the ESI–MS/MS spectra showed different fragment ions that indicated the presence of nine methylenes between a and b, one methylene between b and c’, and one methylene between c’ and d (Appendix A).

Osirisyne I (**3**) was obtained as a white amorphous solid. The molecular formula, C_47_H_72_O_11_, was established from a HRESIMS molecular ion peak at *m*/*z* 811.4998 [M − H]^−^, indicating 12 degrees of unsaturation (Appendix A). Analysis of the 1D and 2D ^1^H and ^13^C NMR data for **3** (Table 2, Appendix A) revealed resonances and correlations consistent with those of a long-chain highly oxygenated polyacetylene, like osirisyne G (**1**). The ^1^H NMR spectrum of **3** recorded in CD_3_OD showed the presence of six olefinic protons (δ_H_ 5.88 (1H, ddd, *J* = 15.3, 6.2, 1.2 Hz), δ_H_ 5.76 (1H, ddd, *J* = 15.4, 5.7, 1.1 Hz), δ_H_ 5.62 (1H, dq, *J* = 14.5, 7.0 Hz), δ_H_ 5.62 (1H, dq, *J* = 14.5, 7.0 Hz), δ_H_ 5.43 (1H, ddt, 15.3, 7.1, 1.3), and δ_H_ 5.43 (1H, ddt, 15.3, 7.1, 1.3), an acetylenic proton (δ_H_ 2.91 (1H, d, *J* = 2.2 Hz), nine oxygenated methines (δ_H_ 5.11 (1H, q, *J* = 1.7 Hz), δ_H_ 4.82 (1H, dm, *J* = 5.7 Hz), δ_H_ 4.33 (1H, td, *J* = 6.7, 1.6 Hz), δ_H_ 4.19 (1H, d, *J* = 5.2 Hz), δ_H_ 4.09 (1H, q, *J* = 6.1 Hz), δ_H_ 3.98 (1H, q, 6.3 Hz), δ_H_ 3.97 (1H, q, 6.3 Hz), δ_H_ 3.69 (1H, m), and δ_H_ 3.56 (1H, ddd, *J* = 8.6, 5.0, 3.4 Hz), and a series of methylene groups in the range δ_H_ 2.50–1.30. The ^13^C NMR spectrum of **3** showed the presence of a carboxylic acid C-1 (δ_C_ 161.5), eight sp carbon due to four triple bonds C-2, C-3, C-32, C-33, C-35, C-36, C-46, and C-47 (δ_C_ 80.7, 83.3, 85.5, 84.4, 80.7, 82.1, 84.4, 74.9), six sp^2^ carbons due to two double bonds C-19, C-20, C-25, C-26, C-43, and C-44 (δ_C_ 132.5, 134.5, 132.3, 134.4 136.2, 130.5), nine oxymethines C-4, C-5, C-21, C-27, C-31, C-34, C-38, C-42, and C-45 (δ_C_ 67.2, 75.5, 73.6, 73.6, 62.6, 52.5, 70.9, 72.4, 62.6), and several methylene groups were also present. Osirisyne I (**3**) is similar to osirisyne G (**1**), however, the ^13^C NMR spectrum of **3** indicated the disappearance of the carbonyl group C-19. Instead, signals for a new double bond, C-19, C-20, and a hydroxyl-bearing methine, C-21, appeared. This partial structure was confirmed by COSY correlations between H-18 and H-19; H-19 and H-20; H-20 and H-21; and between H-21 and H-22. Osirisyne I (**3**) was also different from osirisyne G (**1**) due to the presence of the methylene C-6 (δ_H_ 1.47 (1H, m), 1.67 (1H, m); δ_C_ 33.3) instead of an oxygenated methine. The molecular formula determined by HRESIMS, C_47_H_72_O_11_ for osirisyne I (**3**) and C_47_H_72_O_12_ for osirisyne G (**1**), in addition to the HSQC correlation between H-6 and C-6 and the HMBC correlations between H-6, C-4, and C-5, confirmed this difference. The connectivities between partial structures a’, b’, c, and d for **3** as well as the number of the linking methylene groups were established on the basis of the molecular formula and ESI–MS/MS data. Regarding osirisyne G (**1**), the ESI–MS/MS spectra showed different fragment ions that indicated the presence of nine methylenes between a’ and b’, one methylene between b’ and c, and one methylene between c and d (Appendix A).

### 2.2. Biological Activity

While the CH_2_Cl_2_-MeOH extract from *Haliclona* sp. sponge presented significant anti-tyrosinase activity (31.1%) (Figure 6), one of these molecules, osirisyne E (**6**), was already described as an enzyme inhibitor. This molecule, as well as osyrisines C and F, had shown Na^+^/K^+^ ATPase and reverse transcriptase (RT) inhibitory activities [13]. The six isolated osirisynes (**1**–**6**) were submitted to a biological evaluation against seven different targets involved in aging or age-related diseases. These targets include biological assays on catalase and sirtuin 1 activation and on CDK7, Fyn kinase, tyrosinase, elastase, and proteasome inhibition.

Catalase is a common enzyme located at the peroxisome that prevents cell oxidative damage (oxidized proteins, lipids, and DNA) by converting hydrogen peroxide (H_2_O_2_) into water (H_2_O) and dioxygen (O_2_). This antioxidant enzyme prevents the accumulation of hydrogen peroxide, which is continuously produced by metabolic reactions and belongs to the reactive oxygen species (ROS), in cellular organelles and tissues. Indeed, ROS are associated to the pathogenesis of numerous diseases including age-related diseases [23,24,25]. Finding natural products that are catalase activators can increased the intracellular antioxidant defense system capacity and can be useful in preventing these diseases.

Sirtuin 1 is a member of the sirtuin family of proteins, a group of very promising targets for anti-aging approaches [26] with activities linked to crucial biological processes like regulating ribosomal DNA recombination, gene silencing, DNA repair, chromosomal stability, and longevity [27].

CDK7 is one of the cyclin-dependent kinases (CDKs), known for their critical roles in cell cycle regulation but also involved in other physiological process like DNA repair and transcription [28]. This kinase has been reported in recent studies to be crucial for the pathogenesis of certain cancer types driven by transcription of a key set of genes and has been validated as a therapeutic target for cancers [29,30,31].

Fyn is a member of the Src family of protein tyrosine kinases (PTKs), an important class of molecules in human biology. Fyn’s biological functions are diverse, and include signaling via the T cell receptor, regulation of brain function, as well as adhesion mediated signaling [32]. Recent studies highlight the involvement of this kinase in different age-related diseases such as cancers [33] or Alzheimer’s disease [34]. Fyn interacts with both protein Tau and amyloid β-peptide, two key players responsible for the major pathologic hallmarks of Alzheimer’s disease [35,36] and inhibitors of this kinase seem to be a promising novel approach therapy of this disease [18].

Tyrosinase, a copper-containing metalloenzyme, is a key enzyme involved in melanogenic processes [37]. In humans, melanin helps defend skin from the damage caused by UV light, however, excess levels of melanin can cause various dermatological disorders including hyperpigmentations, melisma, freckles, and age spots. Many tyrosinase inhibitors have been used in cosmetics and pharmaceutical products for the prevention of overproduction of melanin in the epidermis, however, side effects may occur following the chronic exposure to these compounds, so research of new tyrosinase inhibitors is still important for the development of new cosmeceuticals agents [38].

Elastase is a proteinase enzyme capable of degrading elastin, the main component of the elastic fibers responsible for the mechanical properties of connective tissue [39,40]. In the skin, the elastic fibers, together with the collagenous fibers, form a network under the epidermis and are responsible for skin elasticity. Therefore, elastase, by breaking down elastin, decreases the skin elasticity and increases skin aging, resulting in visible skin changes like wrinkles. So, elastase inhibitors are important cosmeceuticals agents by preventing loss of skin elasticity.

Proteasomes are protein complexes containing a common core, referred to as the 20S proteasome, that degrade unneeded or damaged proteins by proteolysis, this process is often called the ubiquitin-proteasome pathway [41]. This ubiquitin-proteasome pathway may be critical in cell cycle regulation, and due to these multiple functions, proteasome malfunctions are involved in a certain number of pathologies, in particular those linked to aging such as cancers and neurodegenerative diseases. Therefore, inhibitors of proteasome are highly sought for the treatment of these diseases [42].

Five compounds (**1**; **3**–**6**) inhibited proteasome activity and two compounds (**5**–**6**) inhibited CDK7 and Fyn kinase (Table 3). Osirisyne B (**5**) was the most active compound with IC_50_ on FynB kinase, CDK7 kinase, and proteasome inhibition of 18.44 µM, 9.13 µM, and 0.26 µM, respectively.

The activity of these compounds seems to be related to the number of oxygen atoms. Osirisynes B (**5**) and E (**6**) are the two compounds with the lowest number of oxygens and presented inhibition activities on three enzymes, FYN kinase, proteasome, and CDK7. Furthermore, comparison between osirisyne B (**5**) and osirisyne E (**6**) highlighted the increase of the inhibition activity with the presence of a ketone instead of a hydroxyl function. In our continuing search for bioactive metabolites, studies on this family of compounds from sponges are still going on and a structure–activity relationship (SAR) study will be made when more similar compounds are isolated and tested.

## 3. Materials and Methods

### 3.1. General Experiment Procedures

Optical rotations were measured on an MCP 200 Anton Paar modular circular polarimeter at 25 °C (MeOH, *c* in g/100 mL) in a 10 × 5 mm i.d., 0.2 mL sample cell. ^1^H and ^13^C NMR data were acquired with a Bruker Avance III–600 MHz spectrometer equipped with a 5 mm TCI Cryoprobe in 1.7 mm o.d. capillary tube at 300 K. Chemical shifts were referenced using the corresponding solvent signals (δ_H_ 3.31 and δ_C_ 49.00 for CD_3_OD 99.96%-d). The spectra were processed using 1D and 2D NMR MNova software.

HRESIMS and MS/MS spectra were recorded using a Waters SYNAPT G2 HDMS mass spectrometer with an API source. For MS/MS, parameters of ionization were: ESI^+^: capillary voltage: 2.8 kV, cone voltage: 20 V, gas flow (N_2_): 100 L/h, collision gas: Ar.ESI^−^: capillary voltage: −2.27 kV, cone voltage: −20 V, gas flow (N_2_): 100 L/h, collision gas: Ar.

Samples were solubilized in 300 µL of MeOH and then diluted to 1/10 in a solution of 1 mM CuSO_4_ in MeOH. 

The sponge was lyophilized with Cosmos −80 °C CRYOTEC and extracted with Dionex ASE 300. MPLC separations were carried out on Buchi Sepacore flash systems C-605/C-615/C-660 and a glass column (230 × 15 mm i.d.) packed with Macherey-Nagel MN Kieselgel silica gel (60−200 μm). Precoated TLC sheets of silica gel 60, Alugram SIL G/UV254, were used, and spots were visualized on the basis of the UV absorbance at 254 nm and by heating silica gel plates sprayed with formaldehyde−sulfuric acid or Dragendorff reagents. Analytical HPLC was carried out using a Phenomenex Gemini C_18_ (150 × 4.6 mm i.d., 3 μm) column and was performed on a Thermo Scientific Dionex Ultimate 3000 system equipped with a photodiode array detector and a Corona detector with Chromeleon software. Semi-preparative HPLC was carried out using a Phenomenex Geminin C_18_ (250 × 10 mm i.d., 5 μm) column and was performed on a Thermo Scientific Dionex Ultimate 3000 system equipped with a photodiode array detector. All solvents were analytical or HPLC grade and were used without further purification.

### 3.2. Animal Material

The sponge *Haliclona* sp. (phylum Porifera, class Demospongiae, order Haplosclerida, family Chalinidae) was collected in May 2013 in Mayotte (12°56,388′ S, 45°03,247′ E at 9–18 m depth). One voucher specimen (RMNH POR 8384) was deposited into the Naturalis Biodiversity Center, the Netherlands. Sponge samples were frozen immediately and kept at −20 °C until processed.

### 3.3. Extraction and Isolation

The frozen sponge (7.7 g, dry weight) was chopped into small pieces and extracted by ASE first with water (×1) and then with MeOH/CH_2_Cl_2_ (1:1, v/v) (×2). After evaporating the solvents under reduced pressure, a brown, oily residue (2.28 g) was obtained. The extract was then subjected to fractionation by MPLC over silica gel in a glass column (230 × 15 mm i.d.), eluting with a combination of cyclohexane, AcOEt, CH_2_Cl_2_, and MeOH of increasing polarity (15 mL min^−1^). Twelve fractions were obtained: F1–F2 eluted with cyclohexane-EtOAc (95:5) over 10 min; F3 eluted with cyclohexane-EtOAc (75:25) over 5 min; F4 eluted with cyclohexane-EtOAc (50:50) over 5 min; F5 eluted with cyclohexane-EtOAc (25:75) over 5 min; F6 eluted with EtOAc over 5 min; F7 eluted with CH_2_Cl_2_ over 5 min; F8 eluted with CH_2_Cl_2_-MeOH (75:25) over 5 min; F9 eluted with CH_2_Cl_2_-MeOH (50:50) over 5 min; F10 eluted with CH_2_Cl_2_-MeOH (25:75) over 5 min; and F11-F12 eluted with MeOH over 10 min.

Fraction F9 (507 mg). Separation of only 100 mg of this fraction was performed by semipreparative HPLC (Phenomenex Geminin C_18_ column) 250 × 10 mm i.d., 5 μm., 4.5 mL min^−1^ gradient elution with 50% ACN-H_2_O (0.1% formic acid) over 25 min, then 50% to 100% ACN-H_2_O (0.1% formic acid) over 10 min and 100% CAN (0.1% formic acid) over 15 min; UV 200 nm) to furnish pure compounds **1** (osirisyne G, 1.4 mg), **2** (osirisyne H, 2.8 mg), **3** (osirisyne I, 1.0 mg), **4** (osirisyne A, 2.2 mg), **5** (osirisyne B, 2.5 mg), and **6** (osirisyne E, 2.7 mg). 

*Osirisyne G* (**1**): amorphous solid, ${\left[\alpha \right]}_{D}^{25}$ +59.5 (*c 0.029*, MeOH); ^1^H and ^13^C NMR, see Table 1; HRESIMS *m*/*z* 827.4950 [M − H]^−^ (calcd for C_47_H_71_O_12_^−^, 827.4951).

*Osirisyne H* (**2**): amorphous solid, ${\left[\alpha \right]}_{D}^{25}$ −35.0 (*c 0.029*, MeOH); ^1^H and ^13^C NMR, see Table 1; HRESIMS *m*/*z* 811.5002 [M − H]^−^ (calcd for C_47_H_71_O_11_^−^, 811.5002).

*Osirisyne I* (**3**): amorphous solid, ${\left[\alpha \right]}_{D}^{25}$ +241.0 (*c 0.214*, MeOH); ^1^H and ^13^C NMR, see Table 1; HRESIMS *m*/*z* 811.4998 [M − H]^−^ (calcd for C_47_H_71_O_11_^−^, 811.5002).

### 3.4. Catalase Activation Assays

Catalase activity was measured using The Amplex^®^ Red Catalase Assay Kit according to the manufacturer’s instructions (Thermo fisher Scientific) [43]. Briefly, catalase first reacts with H_2_O_2_ to produce water and oxygen (O_2_). Next, the Amplex Red reagent reacts with any unreacted H_2_O_2_ in the presence of horseradish peroxidase (HRP) to produce the highly fluorescent oxidation product, resorufin. The fluorescence was measured on a Polar Star Omega (BMG Labtech) plate reader. 

### 3.5. Sirtuin 1 Activation Assays

Sirt1 activity was measured using the SIRT1 Fluorometric Drug Discovery Kit according to the manufacturer’s instructions (Enzo Life Sciences). Briefly, this assay uses a small lysine-acetylated peptide, corresponding to K382 of human p53, as a substrate. The lysine residue is deacetylated by SIRT1, and this process is dependent on the addition of exogenous NAD^+^. The assay was carried out at 37 °C using Greiner white, small volume 384-well plates. First, 4 µL of substrate Fluor de Lys (final concentration 25 μM) were mixed with 4 µL of extract previously diluted 1/100 in assay buffer and 2 µL of the enzyme were added. After an incubation of 15 min at 37 °C, 10 µL of Developer 1X solution (composed by buffer, developer 5X, and Nicotinamide 50 mM) was added and incubated for 45 min at 37 °C.

Afterwards, the fluorescence was measured on a Polar Star Omega (BMG Labtech) plate reader. The fluorescence generated is proportional to the quantity of deacetylated Lysine (i.e., corresponding to Lysine 382). All measurements were performed in triplicate and the final DMSO concentration is 0.1 %. SIRT1 inhibitors nicotinamide (2 mM), suramin (100 µM), and sirtinol (100 µM) were used to confirm the specificity of the reaction. Calculation of net fluorescence included the subtraction of a blank consisting of buffer containing no NAD+ and was expressed as a percentage of control. 

### 3.6. CDK7 Inhibition Assays

The assay was carried out at room temperature (22 °C) using Corning white low volume 384-well plates. All measurements were performed in duplicate, and the final DMSO concentration was 3.3%. The assay buffer contained 20 mM Hepes pH 7.0, 150 mM NaCl, and 10 mM MgCl_2_. (Table 4).

For the positive control, 9.5 µL protein dilution (final concentration 300 nM) were mixed with 0.5 µL of compound dilution ranging from 990–0.006 μM (assay end conc. 33–0.0002 μM). For the selected hits, 9.5 µL protein dilution (final concentration 300 nM) were mixed with 0.5 µL of compound dilution ranging from 1–0.0005 mg/mL (assay end conc. 33–0.016 μg/mL) and preincubated for 120 min.

Then, 5 µL substrate/ATP mix was added (final concentration substrate 30 µM and ATP 125 µM) and the assay was incubated for another 2 h. Afterwards, 5 µL of the 15 µL assay reaction were transferred to new wells and 5 μL ADP Glo Reagent was added to terminate the kinase reaction and deplete the remaining ATP. After 40 min, 10 µL of Kinase Detection Reagent was added to convert ADP to ATP and allow the newly synthesized ATP to be measured using a luciferase/luciferin reaction. After another 40 min, the assay plates were measured in Luminescence mode on a Tecan M1000 plate reader.

The light generated, i.e., luminescent signal, is proportional to the ADP concentration produced and is correlated with kinase activity. The IC_50_ was calculated using XLfit.

In general, it can be stated that the ADP-Glo Kinase Assay worked properly, as can be seen for the control compound Staurosporine, which was run in parallel to the assay.

### 3.7. Proteasome Inhibition Assays

The assay was carried out at room temperature (22 °C) using Corning 4514 black low volume 384-well plates. All measurements were performed in duplicate and the final DMSO concentration was 3.3%. The assay buffer contained 100 mM Tris pH 7.5 and 1 mM MgCl_2_. (Table 5).

For the positive control, 9.5 µL protein dilution (final concentration 9 nM) were mixed with 0.5 µL of compound dilution ranging from 30–0.015 μM (assay end conc. 1000–0.5 nM). For the selected hits, 9.5 µL protein dilution (final concentration 9 nM) were mixed with 0.5 µL of compound dilution ranging from 1–0.0005 mg/mL (assay end conc. 33–0.016 μg/mL) and preincubated for 90 min. Then, 5 µL substrate mix was added (final concentration 5 µM) and incubated for 60 min. The assay plates were measured in Fluorescence mode on a Tecan M1000 plate reader (ex 380 nm, em 460 nm). The IC_50_ was calculated using XLfit.

For assay set-up and validation, the tool compound ONX-0914 was utilized. For this tool compound, a dose-dependent decrease in the Fluorescence signal was observed similar to previous results.

### 3.8. FynB Kinase Inhibition Assays

The assay was carried out at room temperature (22 °C) using Corning 4513 white low volume 384-well plates. All measurements were performed in duplicate and the final DMSO concentration was 3.3 %. The assay buffer contained 20 mM Tris pH 8.0, 170 mM NaCl, and 10 mM MgCl_2_. (Table 6).

For the positive control (Staurosporine), 9.5 µL protein dilution (final concentration 200 nM) were mixed with 0.5 µL of compound dilution ranging from 300–0.15 μM (assay end conc. 10–0.005 μM). For the selected hits, 9.5 µL protein dilution (final concentration 200 nM) were mixed with 0.5 µL of compound dilution ranging from 1–0.0005 mg/mL (assay end conc. 33–0.016 μg/mL) and preincubated for 90 min.

Then, 5 µL substrate/ATP mix was added (final concentration substrate 10 µM and ATP 100 µM) and the assay was incubated for another 2 h. Afterwards, 5 µL of the 15 µL assay reaction were transferred to new wells and 5 µL ADP Glo Reagent was added to terminate the kinase reaction and deplete the remaining ATP. After 40 min, 10 µL of Kinase Detection Reagent was added to convert ADP to ATP and allow the newly synthesized ATP to be measured using a luciferase/luciferin reaction. After another 40 min, the assay plates were measured in Luminescence mode on a Tecan M1000 plate reader. The light generated, i.e., luminescent signal, is proportional to the ADP concentration produced and is correlated with kinase activity. The IC_50_ was calculated using XLfit.

In general, it can be stated that the ADP-Glo Kinase Assay worked properly, as can be seen for the control compound Staurosporine which was run in parallel to the assay.

### 3.9. Tyrosinase Inhibition Assays

The anti-tyrosinase capacity of the extracts or compounds was assayed using an enzymatic method as previously described [45] with minor modifications. Briefly, in a 96-well microplate, 1/15 M PBS (pH 6.8), the tested samples, and 92 U/mL of mushroom tyrosinase (Sigma-Aldrich) were mixed and incubated for 10 min at room temperature, avoiding light exposure. Following the addition of 2.5 mM L-DOPA (Sigma-Aldrich), the mixture was incubated at 25 °C for 5 min. The absorbance at 475 nm of each well was measured using reader Infinite 200 PRO series (Tecan Trading AG, Switzerland) and the MagellanTM software. The percentage inhibition of the tyrosinase activity was calculated by the following equation: [(A − B) − (C − D)]/(A − B) × 100, where A is the control (w/o sample), B is the blank (w/o sample, w/o tyrosinase), C is the sample, and D is the blank sample (w/o tyrosinase). Blanks contained all the aforementioned components except for the enzyme. Kojic acid (KA) and the methanolic extract from the root of Glycyrrhiza glabra L. (Gly) were used as positive controls at final concentrations of 2 and 5 µg/mL, respectively.

### 3.10. Elastase Inhibition Assays

Elastase enzyme activity was evaluated using elastase from porcine pancreas (PPE) type IV and N-succinyl-Ala-Ala-Ala-p-nitroanilide as substrates, as previously described [46]. The amount of released p-nitroaniline, which was hydrolyzed by elastase, was measured spectrophotometrically at 405 nm. The reaction mix was constituted of 70 µL Trizma-base buffer (50 mM, pH 7.5), 10 µL of the extract tested (the final concentration of the extracts was 100 µg/mL), and 5 µL of PPE (0.4725 U/mL), in a 96-well microplate. The samples were incubated for 15 min at room temperature, avoiding light exposure. Subsequently, 15 µL from 0.903 mg/mL N-succinyl-Ala-Ala-Ala-p-nitroanilide were added and the samples were incubated at 37 °C for 30 min. Then, the absorbance of p-nitroaniline production was measured in the reader Infinite 200 PRO series (Tecan). Elastatinal was used as positive control. Experiments were performed in duplicate. The reagents of the assay were purchased from Sigma-Aldrich. The percentage of elastase inhibition was calculated as follows:Inhibition (%) = {[(Abs control − Abs control’s blank) − (Abs sample − Abs sample’s blank)]/[(Abs control − Abs control’s blank)]} × 100(1)
where Abs control is the absorbance of the elastase in the Trizma base buffer, sample solvent, and substrate, and Abs sample is the absorbance of the elastase in the Trizma base buffer, extract or elastatinal, and substrate. Blank experiments were performed for each sample with all the reagents except the enzyme.

## 4. Conclusions

In conclusion, three new long-chain highly oxygenated polyacetylenes, osirisynes G-I (**1**–**3**), were isolated from *Haliclona* sp. together with three known long-chain highly oxygenated polyacetylenes osirisynes A (**4**), B (**5**), and E (**6**). The CH2Cl2-MeOH extract from *Haliclona* sp. sponge presented significant anti-tyrosinase activity (31.1%), however none of the *Haliclona* sp. extracts affected elastase activity significantly. Furthermore, five compounds (**1; 3–6**) showed interesting activities on three different biological assays: CDK7, proteasome, and Fyn kinase inhibition. Osirisyne B (**5**) was the most active compound with IC_50_ on FYNB kinase, CDK7 kinase, and proteasome inhibition of 18.44 µM, 9.13 µM, and 0.26 µM, respectively. This molecule differed from osirisyne E (**6**) only by the presence of a ketone (C-19) instead of an allylic oxymethine and differed from osirisynes A (**4**), G (**1**), and H (**2**) by a lower number of oxymethines, so structure-activity relationship studies between osirisyne B (**5**) and these three enzymes could be undertaken in order to develop more selective and stable analogues for their therapeutic potential.

## Figures and Tables

**Figure 1 marinedrugs-18-00350-f001:**
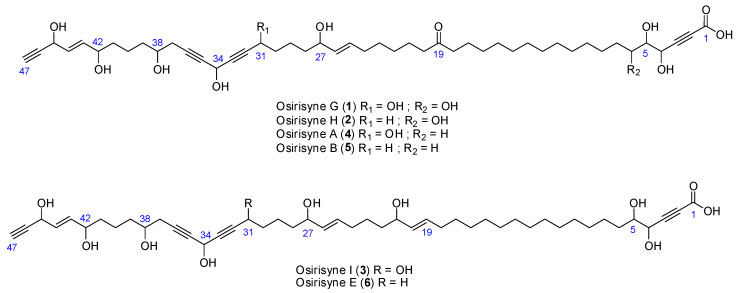
Chemical structures of compounds **1**–**6**.

**Figure 2 marinedrugs-18-00350-f002:**
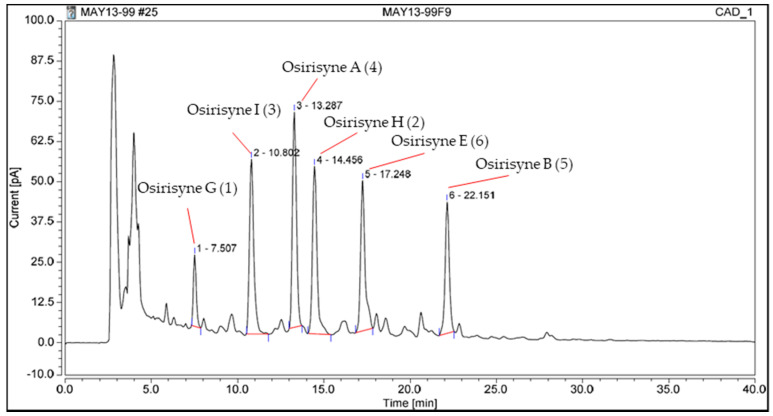
Chromatogram of the fraction worked in semipreparative HPLC with molecules associated to the peaks.

**Figure 3 marinedrugs-18-00350-f003:**
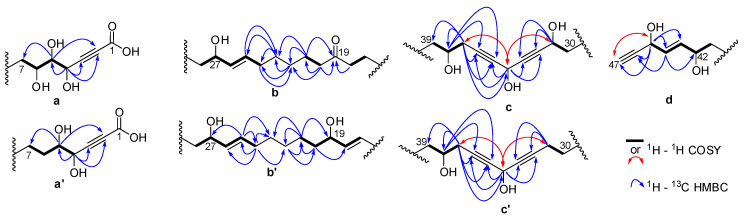
Partial structures of osirisynes G-I (**1**–**3**) with key ^1^H–^1^H COSY and ^1^H–^13^C HMBC correlations.

**Figure 4 marinedrugs-18-00350-f004:**
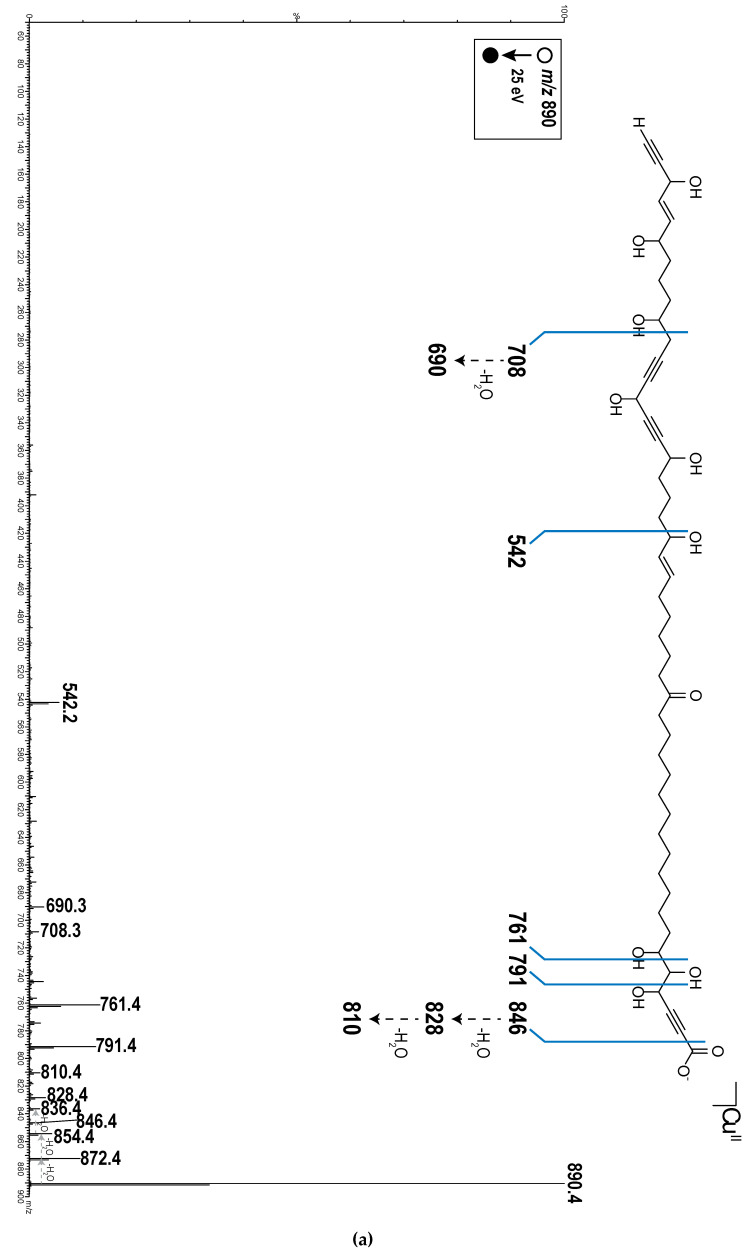
ESI^+^–MS/MS (**a**) and ESI^−^–MS/MS (**b**) spectra of osirisyne G (**1**) with outlines of dissociation of the precursor ions.

**Figure 5 marinedrugs-18-00350-f005:**
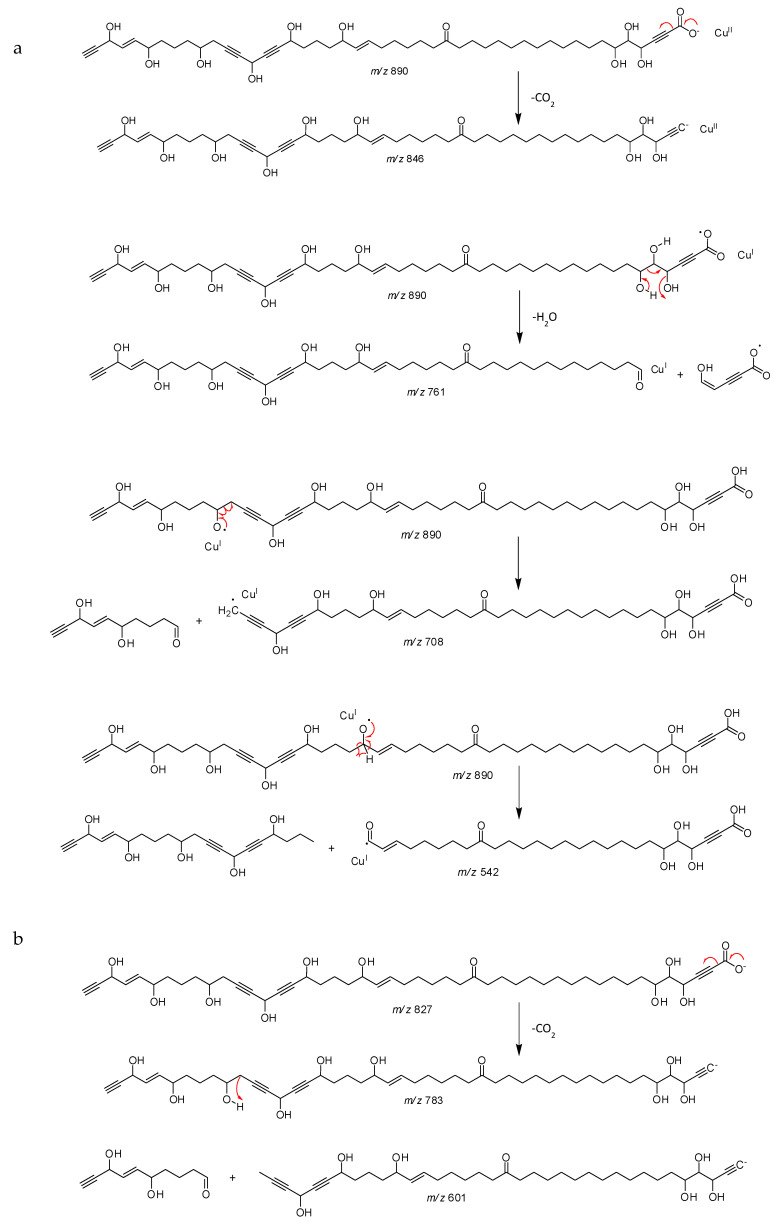
Dissociation mechanisms of the fragmentation of osirisyne G (**1**) in ESI^+^–MS/MS (**a**) and ESI^−^–MS/MS (**b**) with the mass *m*/*z* of the different fragments.

**Figure 6 marinedrugs-18-00350-f006:**
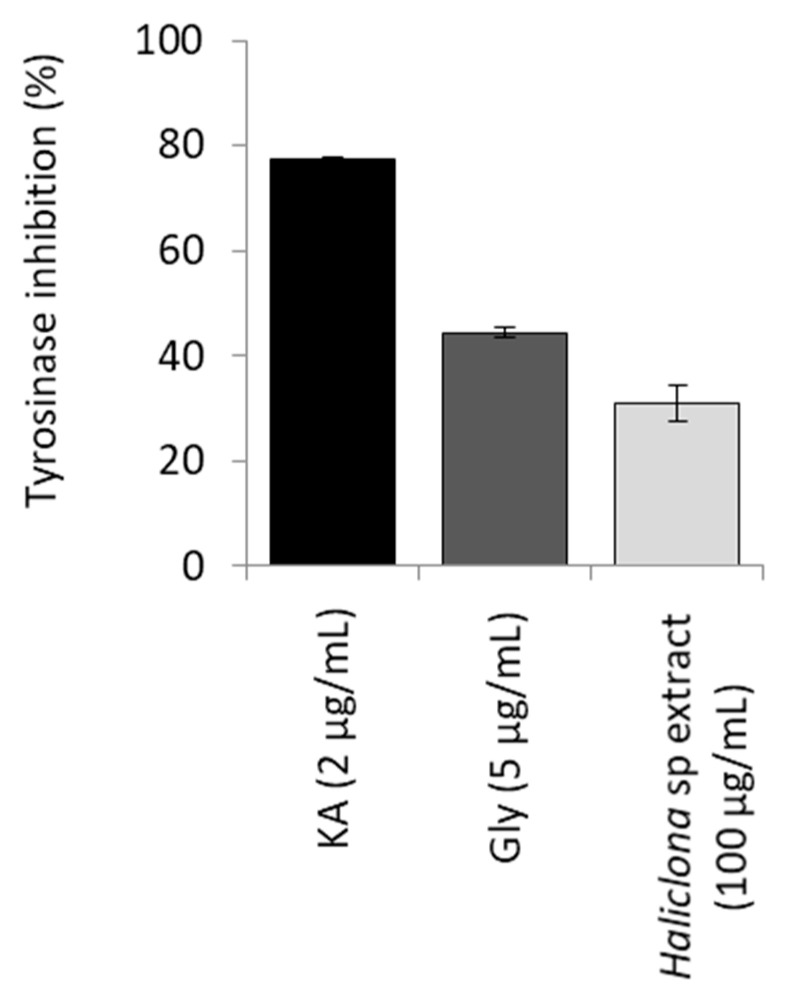
The CH_2_Cl_2_-MeOH extract of the lyophilized sponge *Haliclona* sp. exerted anti-melanogenic properties. Bars, ±SD; n ≥ 2.

**Table 1 marinedrugs-18-00350-t001:** 1D and 2D NMR spectroscopic data (^1^H 600 MHz, ^13^C 150 MHz, CD_3_OD) for osirisyne G (**1**) at 300 K.

No.	δ_C_, Type	δ_H_, mult. (*J* in Hz)	COSY (^1^H-^1^H)	HMBC (^1^H-^13^C)
1	161.3, C			
2	79.9, C			
3	83.4, C			
4	65.0, CH	4.60, 1H, d (4.2)	5	2, 3, 5
5	78.5, CH	3.43, 1H, dd (8.1, 4.3)	4, 6	2, 4, 6, 7
6	72.7, CH	3.61, 1H, td (8.6, 2.5)	5, 7	4, 8
7	34.3, CH_2_	1.78, 1H, m; 1.38, 1H, m	6, 8	
8	24.0, CH_2_	1.57, 2H, m		
9–16	30.8, CH_2_	1.40–1.30, 16H, m		
17	24.8, CH_2_	1.54, 2H, m	18	
18	43.4, CH_2_	2.46, 2H, t (7.4)	17	16, 17, 19
19	214.5, C			
20	43.5, CH_2_	2.44, 2H, t (8.1)	21	19, 21, 22
21	24.8, CH_2_	1.54, 2H, m	20	19, 20, 22, 23
22	30.2, CH_2_	1.40, 2H, m	21	21, 23, 24
23	30.2, CH_2_	1.48, 1H, m; 1.38, 1H, m	24	21, 22, 24, 25
24	33.1, CH_2_	2.05, 2H, q (7.1)	23, 25	22, 23, 25, 26
25	132.3, CH	5.62, 1H, dtd (15.3, 6.5, 0.8)	24, 26	23, 24, 26, 27
26	134.3, CH	5.43, 1H, ddt (15.3, 7.1, 1.4)	25, 27	24, 27
27	73.5, CH	3.97, 1H, m	26, 28	25, 26, 28, 29
28	38.1, CH_2_	1.56, 1H, m, 1.49, 1H, m	27	27
29	22.4, CH_2_	1.50, 2H, m		27, 31
30	38.5, CH_2_	1.67, 2H, m	31	31, 32
31	62.5, CH	4.33, 1H, td (6.7, 1.6)	30, 34	29, 30, 32, 33
32	85.0, C			
33	83.8, C			
34	52.3, CH	5.11, 1H, m	31, 37	32, 33, 35, 36
35	80.9, C			
36	81.9, C			
37	28.0, CH_2_	2.38, 2H, dd (5.9, 1.9)	34, 38	35, 36, 38, 39
38	70.7, CH	3.69, 1H, tt (10.9, 6.3)	37, 39	36, 37, 39, 40
39	37.0, CH_2_	1.50, 2H, m	38	37, 38
		1.64, 2H, m		
40	22.6, CH_2_	1.49, 2H, m		38, 42
41	38.1 CH_2_	1.58, 1H, m; 1.50, 1H, m	42	42, 43
42	72.2, CH	4.09, 1H, q (6.0)	41, 43	40, 41, 43
43	136.0 CH	5.88, 1H, ddd (15.4, 6.2, 1.3)	42, 44	
44	130.3, CH	5.76, 1H, ddd (15.4, 5.7, 1.1)	43, 45	42, 43, 45, 46
45	62.5, CH	4.82, 1H, dm (5.7)	44, 47	43, 44, 46, 47
46	84.7 C			
47	74.9, CH	2.92, 1H, d (2.2)	45	45

**Table 2 marinedrugs-18-00350-t002:** 1D NMR spectroscopic data (^1^H 600MHz, ^13^C 150 MHz, CD_3_OD) for osirisynes H (**2**) and I (**3**) at 300 K.

No.	Osirisyne H (2)		Osirisyne I (3)	
	δ_C_, Type	δ_H_, mult. (*J* in Hz)	δ_C_, Type	δ_H_, mult. (*J* in Hz)
1	158.6, C		161.5, C	
2	80.0, C		80.7, C	
3	83.3, C		83.3, C	
4	65.0, CH	4.60, 1H, d (4.3)	67.2, CH	4.19, 1H, d (5.2)
5	78.5, CH	3.43, 1H, dd (8.1, 4.3)	75.5, CH	3.56, 1H, ddd (8.6, 5.0, 3.4)
6	72.9, CH	3.62, 1H, td (8.8, 2.3)	33.3, CH_2_	1.67, 1H, m; 1.47, 1H, m
7	34.1, CH_2_	1.77, 1H, m; 1.38, 1H, m	26.9, CH_2_	1.54, 1H, m; 1.35, 1H, m
8	24.7, CH_2_	1.56, 2H, m	30.8, CH_2_	1.40–1.30, 2H, m
9–16	30.8, CH_2_	1.40–1.30, 16H, m	30.8, CH_2_	1.40–1.30, 16H, m
17	24.3, CH_2_	1.54, 2H, m	30.5, CH_2_	1.39, 2H, m
18	43.3, CH_2_	2.45, 2H, t (7,4)	33.3, CH_2_	2.05, 2H, m
19	214.5, C		132.5, CH	5.62, 1H, dq (14.5, 7.0)
20	43.3, CH_2_	2.45, 2H, t (7.4)	134.5, CH	5.43, 1H, ddt (15.3, 7.1, 1.3)
21	24.3, CH_2_	1.54, 2H, m	73.6, CH	3.97, 1H, q (6.3)
22	30.1, CH_2_	1.41, 2H, m	37.9, CH_2_	1.56, 1H, m; 1.45, 1H, m
23	30.1, CH_2_	1.40, 2H, m	26.3, CH_2_	1.39, 2H, m
24	33.0, CH_2_	2.05, 2H, q (7.1)	33.2, CH_2_	2.05, 2H, m
25	132.2, CH	5.61, 1H, dtd (15.3, 6.7, 0.6)	132.3, CH	5.62, 1H, dq (14.5, 7.0)
26	134.3, CH	5.42, 1H, ddt (15.3, 7.1, 1.3)	134.4, CH	5.43, 1H, ddt (15.3, 7.1, 1.3)
27	73.4, CH	3.97, 1H, q (6.3)	73.6, CH	3.98, 1H, q (6.3)
28	37.5, CH_2_	1.51, 2H, m	38.1, CH_2_	1.50, 2H, m
29	26.2, CH_2_	1.43, 2H, m	22.5, CH_2_	1.47, 2H, m
30	29.2, CH_2_	1.52, 2H, m	38.8, CH_2_	1.68, 2H, m
31	19.1, CH_2_	2.23, 2H, td (6.9, 2.0)	62.6, CH	4.33, 1H, td (6.7, 1.6)
32	79.3, C		85.5, C	
33	84.3, C		84.4, C	
34	52.4, CH	5.04, 1H, quint (1.8)	52.5, CH	5.11, 1H, q (1.7)
35	81.1, C		80.7, C	
36	81.5, C		82.1, C	
37	28.0, CH_2_	2.38, 2H, dq (5.8, 1.0)	28.1, CH_2_	2.38, 2H, dd (5.9, 2.0)
38	70.6, CH	3.69, 1H, m	70.9, CH	3.69, 1H, m
39	36.6, CH_2_	1.50, 1H, m; 1.65, 1H, m	37.0, CH_2_	1.50, 1H, m; 1.64, 1H, m
40	22.3, CH_2_	1.49, 2H, m	22.5, CH_2_	1.49, 2H, m
41	37.8 CH_2_	1.58, 1H, m; 1.50, 1H, m	38.1 CH_2_	1.60, 1H, m; 1.51, 1H, m
42	72.2, CH	4.09, 1H, q (6.1)	72.4, CH	4.09, 1H, q (6.1)
43	136.0 CH	5.88, 1H, ddd (15.4, 6.2, 1.3)	136.2 CH	5.88, 1H, ddd (15.3, 6.2, 1.2)
44	130.4, CH	5.76, 1H, ddd (15.4, 5.7, 1.1)	130.5, CH	5.76, 1H, ddd (15.4, 5.7, 1.1)
45	62.4, CH	4.82, 1H, dm (5.7)	62.6, CH	4.82, 1H, dm (5.7)
46	84.0 C		84.4 C	
47	74.5, CH	2.92, 1H, d (2.2)	74.9, CH	2.91, 1H, d (2.2)

**Table 3 marinedrugs-18-00350-t003:** IC_50_ (µM) of the 6 compounds (**1**–**6**) on CDK7, proteasome, and Fyn kinase inhibition.

Compound	FynB Kinase Inhibition	CDK7 Kinase Inhibition	Proteasome Inhibition
Osirisyne G (**1**)	>40.20		>40.20
Osirisyne H (**2**)	>40.99		
Osirisyne I (**3**)	>40.99		>40.99
Osirisyne A (**4**)	>40.99		>40.99
Osirisyne B (**5**)	18.44	9.13	0.26
Osirisyne E (**6**)	>41.81	>41.81	>41.81
Staurosporine	0.137	0.172	
ONX-0914			4.63 × 10^−5^

**Table 4 marinedrugs-18-00350-t004:** ADP-Glo Kinase Assay (Promega) [44].

protein	300 nM CDK7 (Crelux construct CZY-3, PC09891)
substrate	30 μM CDKtide
ATP	125 μM Ultra Pure ATP (Sigma)
buffer	20 mM Hepes pH 7.5, 150 mM NaCl, 10 mM MgCl_2_
extracts	stock conc. 10 mg/mL in DMSO
instrument	TECAN INFINITE M1000 PRO
settings	Mode: Luminescence; Integration time: 100 ms
Compound plates	1 mg/mL in DMSO

**Table 5 marinedrugs-18-00350-t005:** Fluorescence Intensity Assay.

Protein	Yeast proteasome (TUM Groll group)
Storage buffer	20 mM Tris/HCl pH 7.5
substrate	Suc-Leu-Leu-Val-Tyr-AMC (Enzo Lifesciences, BML-P802-0005)
assay buffer	100 mM Tris pH 7.5, 1 mM MgCl_2_
instrument	TECAN INFINITE M1000 PRO
settings	mode: Fluorescence Intensity; ex 380 nm, em 460 nm; bandwidth: +/−10 nm
Compound plates	1 mg/mL in DMSO

**Table 6 marinedrugs-18-00350-t006:** ADP-Glo Kinase Assay (Promega) [44].

protein	200 nM FynB wt (Crelux construct CTX4, PC09815-1)
substrate	10 μM Fyn substrate (Enzo, P-215)
ATP	100 μM Ultra Pure ATP (Promega)
buffer	20 mM Tris pH 8.0, 170 mM NaCl, 10 mM MgCl_2_
instrument	TECAN INFINITE M1000 PRO
settings	Mode: Luminescence; Integration time: 100 ms
Compound plates	1 mg/mL in DMSO

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
