# Peer review of "Osirisynes G-I, New Long-Chain Highly Oxygenated Polyacetylenes from the Mayotte Marine Sponge Haliclona sp."

_marinedrugs, 2020, doi:10.3390/md18070350_

Round 1
Reviewer 1 Report
This paper is related to the isolation and identification of 6 Osirisynes, including three new congeners, from a Mayotte Marine Sponge.
This study represents a strong piece of experimental work including extraction, isolation and characterization using various MS and NMR methods. Several biological tests were conducted but I do not feel qualified to judge this part of the work.
As far as the chemical/analytical part of the study is concerned, I especially appreciated the efforts made by the authors to propose reasonable dissociation pathways to account for the CID data. As numerously stated in many occasions in organic chemistry, a mechanism is the good one until someone proposes a more reasonable one. So, in absence of any theoretical data to support the mechanistic proposal, I consider that the authors made a nice job. I would just suggest, since a broad audience journal is targeted, to add a few details, comments and references regarding the proposal. In particular, the role of the CU2+ ion in the positive ion mode and the involved electron transfer generating a dissociating radical site on the molecule backbone should be somehow referenced and discussed... for non expert reader.
The quality of the MSMS spectra presented in the main text should be increased and there is a small typo on line 215 (ESI-).
The second thing that I would like to discuss is the purity of the compounds. It is stated that the compounds are isolated but, except the NMR spectra, we have to trust the authors. What about also the presence of isomers? Did you run LC-MS experiments? Osirisynes H and A are isomers by the relative positions of the H and OH groups. Maybe some other isomers could be present in quantities below the NMR sensitivity threshold. Also there are numerous stereogenic centers all along the molecule backbone. What about the configurations? What about the presence of enantiomers or diastereoisomers? That should be discussed.
It is also interesting to note that between the two families of molecules, the oxygen atom is present on a different carbon atom (C19 for the C=O and C21 for the C-OH). Is it discussed somewhere in the literature? Is this difference coming from the biosynthesis of these molecules?
I would have appreciated to have a look at LC-MS chromatograms to answer some of the precedent questions. By the way, in the HRMS spectra, there are some additional signals whose intensities are not negligible : contaminations, other congeners, reference ions for the HRMS data? Also, where are the HRMS spectra for the three known molecules?
In conclusion, the presence of the 6 molecules, including three news, is clearly attested by the present study and their structural characterization, except the configurations of the stereogenic centers that are not discussed, are supported by the NMR and MS data. What is missing is the discussion of the presence of (stereo-regio)isomers, even in small concentrations.
Beside the structural characterization, I would also propose to the authors to comment a little bit about the different general questions such as the position of the oxygenated function...
Author Response
This paper is related to the isolation and identification of 6 Osirisynes, including three new congeners, from a Mayotte Marine Sponge.
This study represents a strong piece of experimental work including extraction, isolation and characterization using various MS and NMR methods. Several biological tests were conducted but I do not feel qualified to judge this part of the work.
As far as the chemical/analytical part of the study is concerned, I especially appreciated the efforts made by the authors to propose reasonable dissociation pathways to account for the CID data. As numerously stated in many occasions in organic chemistry, a mechanism is the good one until someone proposes a more reasonable one. So, in absence of any theoretical data to support the mechanistic proposal, I consider that the authors made a nice job.
- I would just suggest, since a broad audience journal is targeted, to add a few details, comments and references regarding the proposal. In particular, the role of the CU2+ ion in the positive ion mode and the involved electron transfer generating a dissociating radical site on the molecule backbone should be somehow referenced and discussed... for non expert reader.
Response: It was added and detailed in the main text.
- The quality of the MSMS spectra presented in the main text should be increased and there is a small typo on line 215 (ESI-).
Response: The figure was enlarged in order to increase the quality of the spectra. Typo on line 215 was corrected.
- The second thing that I would like to discuss is the purity of the compounds. It is stated that the compounds are isolated but, except the NMR spectra, we have to trust the authors. What about also the presence of isomers? Did you run LC-MS experiments?
Response: The HPLC-CAD chromatogram was added in Figure 2 in order to show the separation of these 6 molecules.
- Osirisynes H and A are isomers by the relative positions of the H and OH groups. Maybe some other isomers could be present in quantities below the NMR sensitivity threshold.
Response: Even if osirisynes H and A are structural isomers, these molecules are well separated in HPLC. By the way, we are agree: it is obvious that other isomers could be present in very small amount, below the NMR sensitivity threshold.
- Also there are numerous stereogenic centers all along the molecule backbone. What about the configurations? What about the presence of enantiomers or diastereoisomers? That should be discussed.
Response: The relative configuration of the molecules remained unassigned. Any attempts to obtain suitable derivatives (Mosher’s esters) for a stereochemical analysis were unsuccessful due to the small amount of products and the high number of chiral carbons. Different diastereoisomers should have been visible on NRM spectra. But this is not the case. Moreover, the of the molecules suggested this is not a racemic mixture.
- It is also interesting to note that between the two families of molecules, the oxygen atom is present on a different carbon atom (C19 for the C=O and C21 for the C-OH). Is it discussed somewhere in the literature? Is this difference coming from the biosynthesis of these molecules?
Response: The difference in the position of the oxygen has already been observed in the literature. For example, Osirisyne A (C-19), osirisyne C (C-21), fulvyne B (C-20) or haliclonyne (C-14). There is no study on the biosynthesis of these long chain polyacetylene in the literature, so we can not explain these diferrences.
- I would have appreciated to have a look at LC-MS chromatograms to answer some of the precedent questions. By the way, in the HRMS spectra, there are some additional signals whose intensities are not negligible : contaminations, other congeners, reference ions for the HRMS data?
Response: HRMS ionization was made by Electrospray. This is a competitive method, however the intensity of the signals is not representative of the quantity of products.
- Also, where are the HRMS spectra for the three known molecules?
Response: They were added in the supporting information.
Reviewer 2 Report
A paper entitled “Osirisynes G–I, new long-chain highly oxygenated polyacetylenes from the Mayotte marine sponge Halichona sp.” is submitted to Marine Drugs for further reviewing and publication. Unfortunately, novelty on chemistry for the new isolates osirisynes G (1) and I (2) is not high to attract readers’ interesting. Discussion on the SAR among these isolates (including new and known compounds) is also not provided. Based on these findings, the referee don’t recommend that this submission is acceptable for publication in Marine Drugs with its present form.
Major comments
- In page 3, line 124, “δH04 (1H, p, J = 1.8 Hz)”. What is “p” for coupling pattern?
- In page 4, Figure 2, the 4J-correlation was cited. In general, 4J-correlation in HMBC was not recognized.
- The stereochemistry (relative or absolute) for all isolates is not determined. Why?
- In page 10, the authors have to check the data for HRESIMS, including experimental data and calculated data. All the data were wrong.
Minor comment
- The title for this submission should be revised as “Osirisynes G–I, New Long-chain Highly Oxygenated Polyacetylenes from the Mayotte Marine Sponge Halichona”. The authors have to recheck the text following the format of journal.
- Page 2, line 75, insaturations→unsaturations
- Page 1, line 43, [6, 7, 8]→[6–8].
- The authors have to know what is the difference between “-“ and “–“ used in the text.
- All the δH and δC should be revised as δH and δC, respectively.
Author Response
A paper entitled “Osirisynes G–I, new long-chain highly oxygenated polyacetylenes from the Mayotte marine sponge Halichona sp.” is submitted to Marine Drugs for further reviewing and publication. Unfortunately, novelty on chemistry for the new isolates osirisynes G (1) and I (2) is not high to attract readers’ interesting. Discussion on the SAR among these isolates (including new and known compounds) is also not provided. Based on these findings, the referee don’t recommend that this submission is acceptable for publication in Marine Drugs with its present form.
Major comments
- In page 3, line 124, “δH04 (1H, p, J = 1.8 Hz)”. What is “p” for coupling pattern?
Response: p was used for "pintet". It was changed by "quint" for quintet.
- In page 4, Figure 2, the 4J-correlation was cited. In general, 4J-correlation in HMBC was not recognized.
Response: Herein, the 4J-correlation appears because of the presence of the triple bonds.
- The stereochemistry (relative or absolute) for all isolates is not determined. Why?
Response: The relative configuration of the molecules remained unassigned. Any attemps to obtained suitable derivatives (Mosher's esters) for a stereochemical analysis were unsuccessful due to the small amount of products and the high number of chiral carbons.
- In page 10, the authors have to check the data for HRESIMS, including experimental data and calculated data. All the data were wrong.
Response: The data were corrected.
Minor comment
- The title for this submission should be revised as “Osirisynes G–I, New Long-chain Highly Oxygenated Polyacetylenes from the Mayotte Marine Sponge Halichona”. The authors have to recheck the text following the format of journal.
Response: This has been done.
- Page 2, line 75, insaturations→unsaturations
Response: This has been corrected.
- Page 1, line 43, [6, 7, 8]→[6–8].
Response: This has been corrected.
- The authors have to know what is the difference between “-“ and “–“ used in the text.
Response: small dash was used to give the number of the atom (ex: C-1). Long dash was used to link two atoms (ex C-1--C-2).
- All the δH and δC should be revised as δH and δC, respectively.
Response: This has been corrected.
Reviewer 3 Report
The authors report on the isolation, structural elucidation and biological activities of a series of known and new polyacetylene osirisynes.
The structural characterization is based on extensive NMR and MS analyses, but a detailed description of the results reported in Figure 4 must be added at page 3 .
Using a Bruker 600 Instrument, as indicated in Materials and Methods, the 13CNMR resonance frequency cannot be 300 MHz as reported in the captions of Tables 1 and 2 .
Details on the relevance of the enzymes tested and on their role must be inserted in paragraph 2.2.
The data for the metabolites 1-6 reported in Table 3 must be given in micromolar, in order to have an immediate comparison of the corresponding values and also with the molecules taken as reference.
In figure 5, mL must replace ml, according to the International System of units.
In References, a series of journal abbreviated titles must be introduced.
Author Response
The authors report on the isolation, structural elucidation and biological activities of a series of known and new polyacetylene osirisynes.
- The structural characterization is based on extensive NMR and MS analyses, but a detailed description of the results reported in Figure 4 must be added at page 3 .
Response: A detailed description was reported in the main text.
- Using a Bruker 600 Instrument, as indicated in Materials and Methods, the 13CNMR resonance frequency cannot be 300 MHz as reported in the captions of Tables 1 and 2 .
Response: This has been corrected.
- Details on the relevance of the enzymes tested and on their role must be inserted in paragraph 2.2.
Response: A detailed description was reported in the main text
- The data for the metabolites 1-6 reported in Table 3 must be given in micromolar, in order to have an immediate comparison of the corresponding values and also with the molecules taken as reference.
Response: This has been corrected.
- In figure 5, mL must replace ml, according to the International System of units.
Response:This has been corrected.
- In References, a series of journal abbreviated titles must be introduced.
Response: This has been corrected.
Round 2
Reviewer 2 Report
Although the authors give positive response for most queries. Unfortunately, the stereochemistry for all isolates were not determined (even relative). Discussion on the SAR among these interesting metabolites were also not provided. At this stage, I don’t recommend that this submission is acceptable for publication with its present form.
Author Response
Reviewer 2
Although the authors give positive response for most queries.
Unfortunately, the stereochemistry for all isolates were not determined (even relative).
Response from the authors:
- The relative configuration of the molecules remained unassigned because any attempts to obtain suitable derivatives (Mosher’s esters) for a stereochemical analysis were unsuccessful due to the small amount of products and the high number of chiral carbons. Sui, B. et al. [1] have determined the absolute configuration of petrocortyne A (2 chiral carbons; 4 stereoisomers) by synthesise the four isomers and as they said, it was “not so much because one of the isomers matched the natural product, but because all of the other isomers did not”. Osirisyne G and I possessed 9 chiral carbons so 512 stereoisomers have to be synthesise for each ; Osirisyne H, A and E possessed 8 chiral carbons so 256 stereoisomers have to be synthesise for each; and Osirisyne B possessed 7 chiral carbons so 128 stereoisomers have to be synthesise. It is not an imaginable work.
- Another possibility was to determine the configuration by computational chemistry. By applying the DP4+ probability for the determination of relative configuration[2] and comparison of the ECD spectra for the absolute configuration. For DP4+, all the diastereoisomers (256 for osirisynes G and I; 128 for osirisynes H, A and E and 64 for osirisyne B) have to be modelized. Furthermore, due to the high flexibility of the long chain, thousands of conformers could be present so a high amount of them have to be modelized to be representative of the real spatial structure of the molecule.
- The last possibility was to determine the configuration by crystallography but these compounds don’t crystalized. Moreover, assays of derivatization in order to obtain crystals would be very limited due to the small amount of products.
Reviewer 2
Discussion on the SAR among these interesting metabolites were also not provided.
Response from the authors:
- We are agree, a SAR study on these metabolites could be very interesting but a real SAR study is a research work in its own right. The idea of this article is to publicly share the activities of these compounds which are not already known for an inhibitory potential on FYN, CDK7 and proteasome so that other research teams can work on it and maybe make a real SAR study on them. In our continuing search for bioactives metabolites, studies on this family of compounds from sponges are still going on and a SAR study will be made when more similar compounds will be isolated and tested.
- For the moment, the only hypothesis was the activity is related to the presence of oxygen atoms. Osirisynes B and E are the two compounds with the lowest number of oxygens and have inhibition activities on 3 enzymes, FYN kinase, proteasome and CDK7. Furthermore, if we compare osirisyne B and osirisyne E, the presence of a ketone instead of a hydroxyl function seems to increase the inhibition activity of the molecule.
- A discussion was added in the main text P.11 L.265 to L.272
[1] Sui, B.; Yeh, E. A.-H.; Curran, D. P. Assignment of the Structure of Petrocortyne A by Mixture Syntheses of Four Candidate Stereoisomers. J. Org. Chem. 2010, 75 (9), 2942–2954. https://doi.org/10.1021/jo100115h.
[2] Grimblat, N.; Zanardi, M. M.; Sarotti, A. M. Beyond DP4: An Improved Probability for the Stereochemical Assignment of Isomeric Compounds Using Quantum Chemical Calculations of NMR Shifts. J. Org. Chem. 2015, 80 (24), 12526–12534. https://doi.org/10.1021/acs.joc.5b02396.
Round 3
Reviewer 2 Report
The authors give positive responses for queries and brief discussion SAR was provided. I recommend that this revised submission is acceptable for publication in Marine Drugs as its present form.
Author Response
Dear reviewer.
Thank you for your answer.
Best regards,
Anne Bialecki